# First Record of Corallivorous Nudibranch *Pinufius* (Gastropoda: Nudibranchia) in the South China Sea: A Suspected New Species of *Pinufius* [†]

**Zhiyu Jia [1], Peng Tian [1,2], Wei Wang [1,2], Bingbing Cao [1], Ziqing Xu [1], Jiaguang Xiao [1,2,*] and Wentao Niu [1,2,*]**

[1] Third Institute of Oceanography, Ministry of Natural Resources, Xiamen 361005, China
[2] Nansha Islands Coral Reef Ecosystem National Observation and Research Station, Guangzhou 510000, China
[*] Correspondence: xiaojiaguang@tio.org.cn (J.X.); wentaoniu@tio.org.cn (W.N.)
[†] urn:lsid:zoobank.org:pub:A9778147-079F-41CD-87A1-B11861D903DA.

**Abstract:** A corallivorous nudibranch from the South China Sea reproduced explosively and caused extensive damage to *Porites* in our aquarium. In this study, morphological and molecular analyses of the nudibranch were conducted and described. Morphologically, this nudibranch was nearly consistent with *Pinufius rebus* in its characteristics intermediate between arminids and aeolids. The only detected difference was that the hook-like denticles on the masticatory border of *P. rebus* were absent in this nudibranch. In a molecular analysis, phylogenetic results based on the cytochrome oxidase subunit-I, 16S rRNA, and histone H3 gene sequences showed that this nudibranch and *P. rebus* form a well-supported sister clade under the superfamily Fionoidea, with significant interspecific divergence (0.18). Thus, we presumed that this nudibranch is a new species of *Pinufius*. Our results extend the distribution of *Pinufius* to the South China Sea, support the current taxonomic status of *Pinufius* under the superfamily Fionoidea, and imply that the species composition of *Pinufius* is more complex than previous records. Moreover, as a corallivorous nudibranch, the potential threat of *Pinufius* to coral health cannot be neglected.

**Keywords:** corallivory; sea slug; Pinufiidae; *Porites*; coral health

## 1. Introduction

Coral reefs are among the most biologically diverse, productive, and fragile marine ecosystems on earth [1,2]. Coral reefs worldwide are suffering extensive deterioration due to extreme climates, anthropogenic disturbances, and biological factors such as competition, disease, and predation [3–5]. The crown-of-thorns starfish *Acanthaster* spp. and the muricid gastropods *Drupella* spp. are coral predators notorious for their substantial damage to corals [6,7]. Moreover, some flatworms and nudibranchs are also potential threats to coral reefs [8–10].

Nudibranchia is a diverse (approximately 2540 species) but taxonomically complex order of marine gastropod mollusks, among which there are many controversies on systematical relationships [11]. Initially, the order Nudibranchia was divided into the four suborders: Doridina, Aeolidina, Dendronotina, and Arminina [12]. More recently, Wägele and Willan [13] redistributed the order into two suborders: Anthobranchia and Cladobranchia. Cladobranchia contains seven superfamilies (Aeolidioidea, Arminoidea, Dendronotoidea, Doridoxoidea, Fionoidea, Proctonotoidea, and Tritonioidea) [14]. Among these, the superfamily Fionoidea has the richest variety of families [15]. Fionoidea, meanwhile, is the most controversial taxon, especially regarding disagreements between traditional morphology and emerging molecular identification [16,17].

As a morphologically special family in the superfamily Fionoidea, Pinufiidae, with its monotypic genus *Pinufius*, has only one species *Pinufius rebus*, which is very different from other members of Fionoidea [18]. The position of the anus is similar to that of some

aeolids, but the external shape more resembles arminids; hence, *Pinufius* was initially classified into Arminina [18]. Subsequently, Rudman [19] redescribed the morphological and physiological features and feeding habits of *Pinufius*. According to distribution records in the literature, *Pinufius* inhabits the coral reefs of the Maldives [18], Australia [19], Indonesia [20], and the Philippines [21], and feeds exclusively on *Porites* spp. [22]. However, Rudman [19] did not give an opinion about the taxonomic status of *Pinufius* because of the similarity between *P. rebus* and some aeolids in radula and cerata. In the most recent classification of Gastropoda, Pinufiidae was classified into the superfamily Fionoidea [15].

Similarly, there is no consensus on the phylogenetic position of *Pinufius*. Pinufiidae once clustered with other ceras-bearing arminids including the species of Janolidae, Madrellidae, and Proctonotidae, under the superfamily Metarminoidea [23]. However, Metarminoidea was not named on the basis of existing genus-level taxa and, hence, cannot be valid [24]. In 2010, a phylogenetic analysis of multiple molecular datasets indicated that *Pinufius* clustered closely with *Doto* species of Dotoidae [21]. However, a morphological study showed that Pinufiidae was more closely related to Proctonotidae [11]. Mahguib and Valdés [25] provided a phylogenetic tree in which *P. rebus* clustered closely with *Lomanotus* spp. (Lonanotidae) and *Eubranchus rustyus* (Eubranchidae), and embedded into a large clade with some families of Aeolidioidea. In a study about host shifting, *Pinufius* was nested within a large clade with some *Phestilla* species of the family Trinchesiidae, which also feed exclusively on *Porites* spp. [20].

In China, 203 species were recorded in the order Nudibranchia [26]. Therein, only three coral-eating *Phestilla* (*Phestilla melanobrachia*, *P. goniophaga*, and *P. fuscostriata*) were reported [27–29], and *Pinufius* was never mentioned. In this study, a species of nudibranch was brought into our aquarium with *Porites* samples from Daya Bay in coastal waters of the northern South China Sea. After 2 months, while the population of this nudibranch broke out, they severely damaged these *Porites* samples and never left *Porites* to hurt other corals. According to primary observations under stereomicroscopes, we deduced that this species was *P. rebus*, according to the obligate association with *Porites* spp., as it had the appearance of this species including intermediate morphological features between aeolids and arminids [18]. However, there were significant differences in gene sequences between the nudibranch in this study and *P. rebus* online. In order to accurately identify the species and reveal the phylogenetic relationship with other relevant nudibranchs, we provide a detailed morphological description and comprehensive molecular phylogenetic analysis of this unknown nudibranch. This study is expected to provide a reference for *Pinufius* research and coral reef conservation in China.

## 2. Materials and Methods

### 2.1. Sample Collection

The tested specimens were collected from the surface of *Porites* samples in our aquarium using a pipette. Specimens for molecular analysis were preserved in 95% ethanol; specimens for morphological analysis were preserved in seawater temporarily and 4% formaldehyde. All specimens examined in this study were deposited in the Third Institute of Oceanography, Ministry of Natural Resources, Xiamen, Fujian, China (HYSS01 to HYSS75).

### 2.2. Morphological Analysis

#### 2.2.1. External Morphology

Thirty-five specimens (HYSS06 to HYSS40) were used for external morphological analysis. They were placed in a petri dish with seawater, and magnesium sulfate solution was slowly added dropwise until the animals were anesthetized. External morphological characteristics were examined and photographed using a stereomicroscope (Leica M205FA; Leica Microsystems, Wetzlar, Germany).

Eggs and egg masses were examined and photographed under a biological microscope (Leica DM5000B; Leica Microsystems) and a stereomicroscope (Leica M205FA).

2.2.2. Internal Morphology

Twenty-three specimens (HYSS41 to HYSS63) were used for dissection of the reproductive system under a dissecting microscope (Leica S6D; Leica Microsystems) and a stereomicroscope (Leica M205FA). Identification of organs was aided by the relevant literature [18].

Buccal masses were extracted under a dissecting microscope (Leica S6D) and soaked in 25% hypochlorous acid for 20 min at 24 °C; they were then rinsed in pure water. The radula and jaw were removed and placed on filter paper ($\Phi$ = 5 μm) using forceps. The filter paper was mounted on a stub with liquid nitrogen and examined under a scanning electron microscope (Quanta 450; FEI, Portland, OR, USA).

For histological examination, specimens (HYSS75) were preserved in 4% paraformaldehyde for 24 h and dehydrated in ethanol. Dehydrated specimens were made transparent in the mixed solution of ethanol and xylene (1:1). Transparent specimens were embedded in melted wax and cooled to −20 °C. The wax-embedded specimens were cut into 4 μm slices. These slices were floated on the surface of 40 °C water to remove any wrinkles and dried on glass slides at 60 °C. The sections were cleared in xylene, a mixed solution of ethanol and xylene (1:1), and ethanol successively to remove the wax. Finally, the sections were stained with hematoxylin–eosin stain. Sections were examined under a biological microscope (Leica DM5000B).

*2.3. Molecular Analysis*

2.3.1. DNA Extraction and Sequencing

Four specimens (HYSS02–HYSS05) were used for molecular analyses. The genomic DNA of specimens was extracted using the Ezup Column Animal Genomic DNA Purification Kit (Sangon Biotech, Shanghai, China). DNA integrity was checked using electrophoresis on 1% agarose gel. Polymerase chain reactions (PCRs) were conducted to amplify the cytochrome oxidase I (COI), 16S ribosomal RNA (16S), and histone H3 gene sequences using a Veriti 96-Well Thermal Cycler (Thermo Fisher Scientific, St. Louis, MO, USA). The primers were HCO2198/LCO1498 for COI [30], 16SsarL/16SR for 16S [31,32], and H3AF/H3AR for histone H3 gene [33]. PCR products were purified with SanPrep Column DNA Gel Extraction Kit (Sangon Biotech) and sequenced using an ABI 3730XL Genetic Analyzer (Sangon Biotech).

2.3.2. Phylogenetic Analysis

Phylogenetic topology was constructed on the basis of concatenated COI–16S–histone H3 sequences of our specimens and 49 other nudibranch species (Supplementary Table S1). Among these species, 38 species belonged to the superfamily Fionoidea, including the genus *Phestilla* whose feeding habits and behaviors were similar to *Pinufius*; 10 species in the superfamily Poctonotoidea were morphologically similar to *Pinufius*. The best partition schemes and substitution models were estimated using a comparison of Akaike information criterion (AIC) scores with jModelTest v2.1.7 [34]. A Bayesian inference (BI) phylogenetic tree was constructed using Mrbayes v3.12 [35]; a set optimal model strategy was selected for different positions (GTR + I + G for COI and 16S and GTR + G for histone H3). Analyses were run for 10,000,000 generations with the Markov chains being sampled every 1000 generations. We determined the burn-in value of the first 2500 trees (25%), and the 50% majority-rule consensus tree was estimated. Pairwise genetic distances based on the COI gene were generated using MEGA X [36].

**3. Results**

*3.1. External Morphology*

Mature specimens were 3–6 mm in length (Figure 1). Body color was usually light brown and changed with the color of hosts. The oral veil was round, about one-fourth the length of the body, slightly exceeded the anterior edge of dorsum, and contained brown speckling, without oral tentacles. The rhinophores were smooth and nonretractile, without

a rhinophoral sheath. There was a white patch between the bases of rhinophores. The dorsum was flat and broad, with many small brown tubercles. When resting, the body was oval, and the foot and partial oral veil were shielded by the dorsum; when crawling, the body was elongated, and the foot usually extended behind the dorsum.

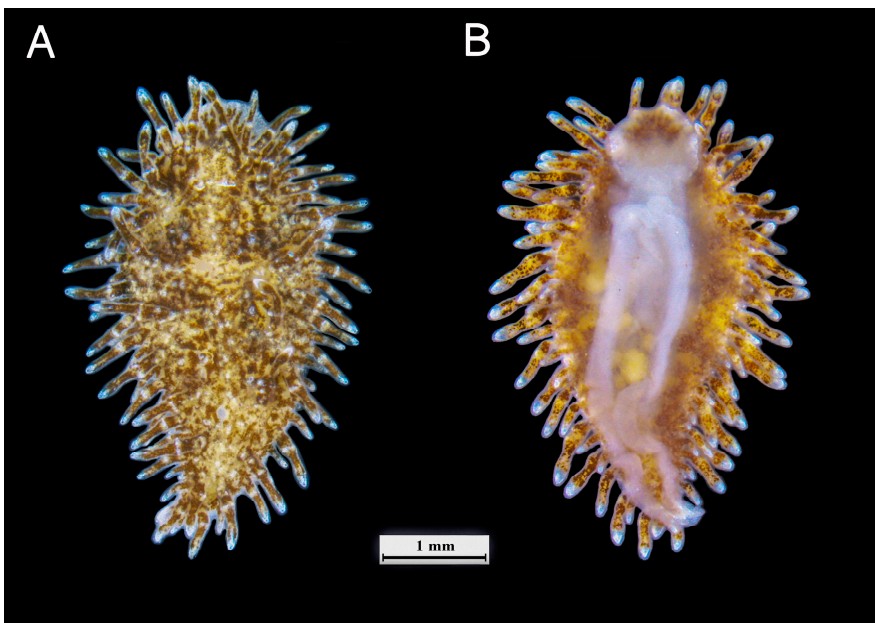

**Figure 1.** The external morphology: (**A**) dorsal view; (**B**) ventral view. Scales: 1 mm.

The cerata were variable in shape (Figure 2A). On the top of dorsum, they were rod-like and partly swollen, arranged symmetrically in five rows. Each row consisted of two pairs of cerata; the pair on the outside (Figure 2A: ot) was smaller than the pair on the inside (Figure 2A: it). Differing from the cerata on the top, the cerata around the edge of dorsum were irregular and dense in a double-row arrangement. The longest inner cerata around the edge of dorsum were on the anterior and usually misidentified as oral tentacles (Figure 2A: ie). The outside cerata around the edge of the dorsum were digitiform, as well as shorter and denser than the inner cerata (Figure 2A: oe). There were 1–2 ceras-like processes between the outside cerata around the edge of dorsum (Figure 2A: cp). The type and arrangement of the cerata in larvae were significantly different from those of adults (Figure 2B).

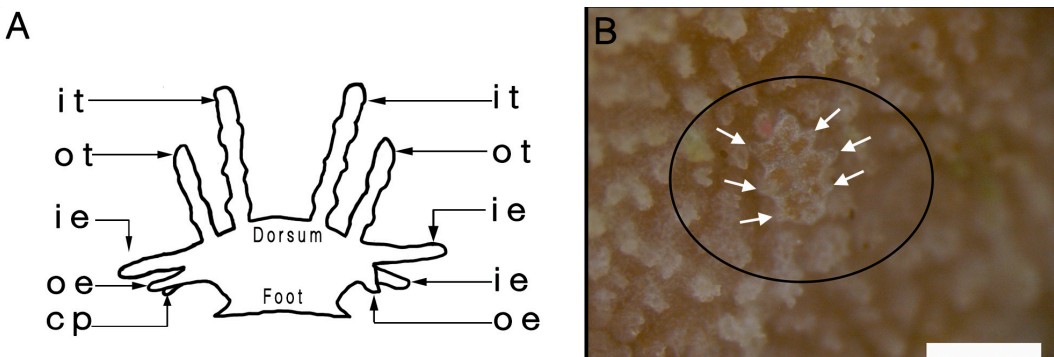

**Figure 2.** The arrangement of cerata. (**A**) The cross-sectional sketch of cerata: it, the inner ceras on the top; ot, the outer ceras on the top; ie, the inner ceras on the edge; oe, the outer ceras on the edge; cp, ceras-like process. (**B**) The larva with limited cerata (white arrow) under the microscope, scale: 500 μm.

The anus was located on the right side of the midline between the third and fourth rows of cerata. The reproductive opening was located on the right side of the foot.

The egg masses were transparent belts (Figure 3A), with hundreds of eggs (Figure 3B). The eggs were 0.2–0.3 mm with transparent membranes (Figure 3B).

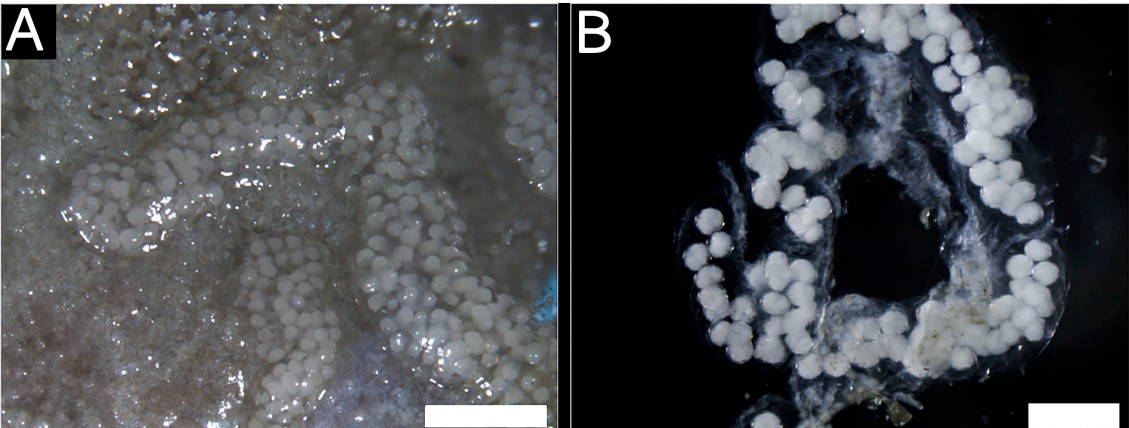

**Figure 3.** Egg masses under microscope: (**A**) Egg masses on the *Porites* sp.; (**B**) hundreds of eggs in transparent belts. Scales: (**A**,**B**) 1 mm.

*3.2. Internal Morphology*

3.2.1. Radula and Jaw

In the buccal mass, a radula was enfolded in a pair of jaws. The radula formula was 17 × 0.1.0 (Figure 4A). The central cusp was strong (Figure 4A: cc), with 6–8 slenderer and shorter primary denticles on each side (Figure 4A: pd). Compared with the other primary denticles, the two pairs closet to central denticle looked runtish. Between the primary denticles, 1–2 secondary denticles were present (Figure 4A: sd). The jaw plates were transparent and triangular (Figure 4B,C). The masticatory border was rough and thick (Figure 4C), without the hook-like denticle of *P. rebus* as Marcus et al. [18] and Rudman [19] described.

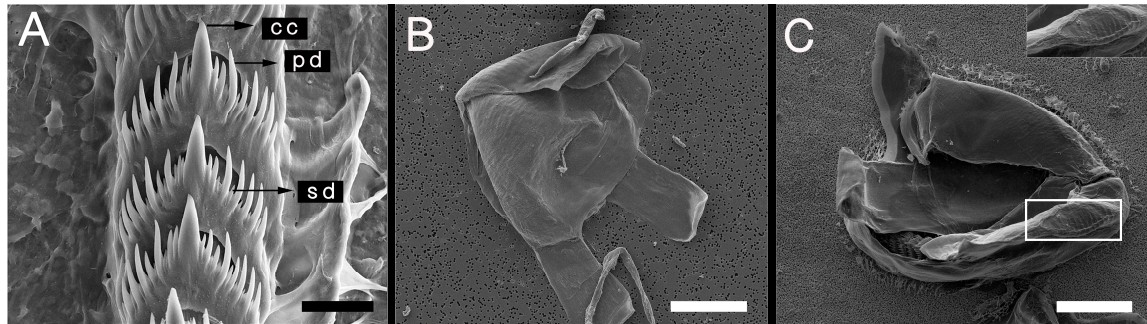

**Figure 4.** Radula and jaw under SEM. (**A**) Radula: cc, central cusp; pd, primary denticle; sd, secondary denticle. (**B**) The outside of jaw plate. (**C**) The inner side of jaw plate with a section of the masticatory border enlarged in the inset. Scales: (**A**,**C**) 100 μm; (**B**) 300 μm.

3.2.2. Reproductive System

The reproductive system (Figure 5) was diaulic, mainly consisting of a penis, a female gland mass, an ampulla, and a bursa copulatrix, as Marcus et al. [18] described (Figure 5B). The penis was slender and connected to an irregular, massive prostate (Figure 5A: p, pr). The bursa copulatrix with a transparent membrane was spherical and connected to an elongated vagina (Figure 5C: bc, v). The ampulla was slightly swollen (Figure 5C: am). The

female gland mass was irregularly cylindrical (Figure 5D: fgm). Many follicles were visible in the posterior of the body through the ventral epidermis (Figure 5E: fo).

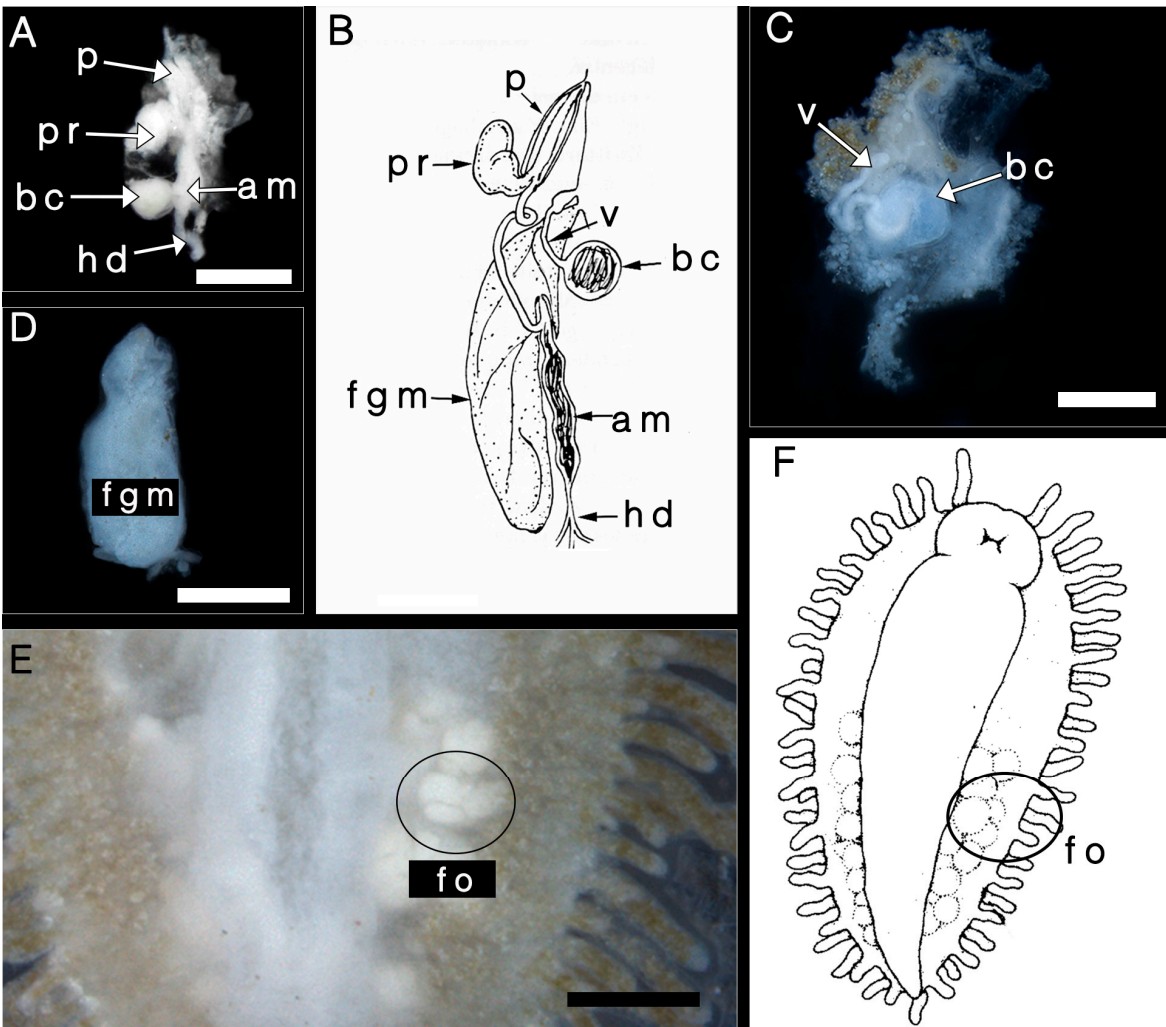

**Figure 5.** Reproductive system. Comparison of digital photos (**A**,**C**–**E**) and drawings by Marcus et al. [18] (**B**) and Rudman [19] (**F**): p, penis; pr, prostate; v, vagina; bc, bursa copulatrix; hd, hermaphroditic duct; am, ampulla; fo, follicle; fgm, female gland mass. Scales: (**A**,**C**,**D**) 500 μm; (**E**) 100 μm.

### 3.2.3. Histological Characteristics

The dorsal epidermis was thin and semitransparent (Figure 6A: de). Under the epidermis, zooxanthellae (Figure 6A: z) were mainly concentrated in the tubercles (Figure 6A: t) and digestive glands (Figure 6A: dg). The digestive gland extended to the cerata, with zooxanthellae (Figure 6B: z). There was no nematocyst in the apex of the cerata (Figure 6B). The epidermic cells of the foot (Figure 6C: ef) were ciliated and columnar (Figure 6C: c). The inner tissue (Figure 6C: it) was large and loose, and a narrow duct (Figure 6C: d) extended out to the epidermis. We believe this to be the foot gland described by Marcus et al. [18]. The connective tissue between the foot and cavity was filled with fibers (Figure 6C: f). The inner structure of the follicles (Figure 6D: fo) was multivesicular.

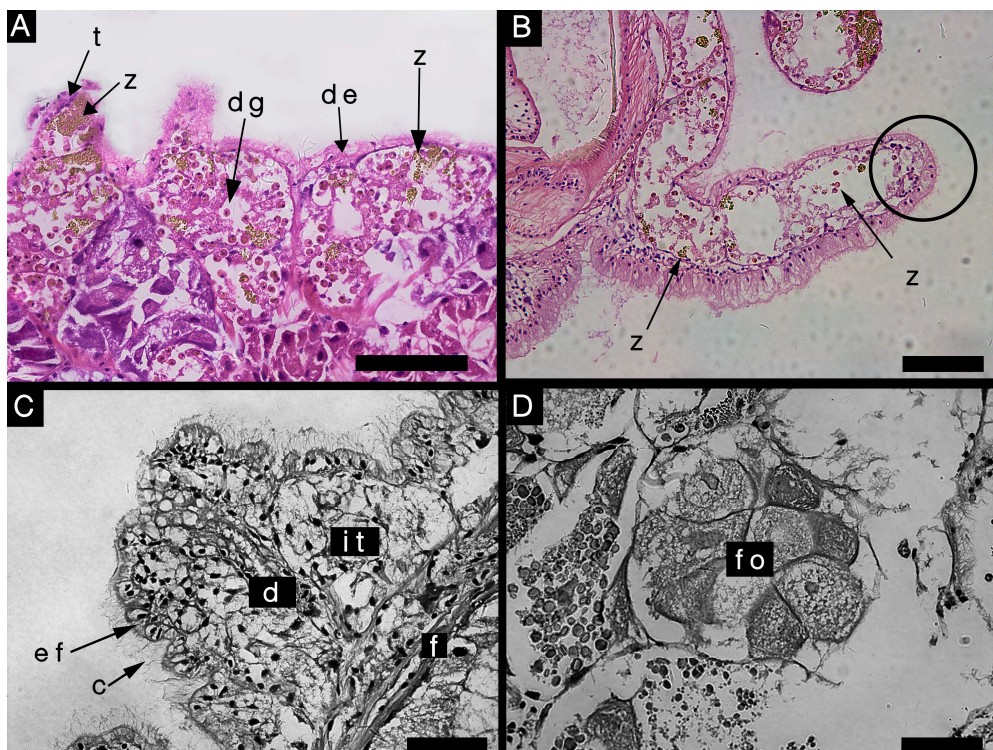

**Figure 6.** Histological transections. (**A**) The dorsum: t, tubercle; z, zooxanthellae; dg, digestive gland; de, the dorsal epidermis. (**B**) Ceras: z, zooxanthellae. (**C**) The foot gland: ef, the epidermis of the foot; c, cilia; d, duct; it, the inner tissue of the foot; f, the fibers of connective tissue. (**D**) Follicle: fo, follicle. Scales: (**A**,**B**) 100 μm, (**C**) 70 μm, (**D**) 50 μm.

### 3.3. Phylogenetic Relationships

After alignment, we obtained three gene matrices including a 658 bp COI fragment, a 452 bp 16S rRNA fragment, and a 328 bp histone H3 fragment. The BI tree based on concatenated gene sequences indicated that all of the interspecific nodes were robust, with strong posterior probabilities (Figure 7). The resultant topology supported that the superfamilies Fionoidea and Proctonotidea formed monophyletic groups. With the exception of *Phestilla sibogae*, all other species of *Phestilla* clustered into a robust branch. The sequences of *P. sibogae* in this study were from Cella et al. [16], and its separation from other *Phestilla* spp. also happened in the studies of Hu et al. [27,28] and Ekimova et al. [37]. Our four nudibranch specimens (HYSS02-05) formed a well-supported monophyletic clade which was the sister species of *Pinufius rebus*. They were nested within the *Phestilla* clade of the superfamily Fionoidea, which had a significant phylogenetic distance from the superfamily Proctontidea.

Furthermore, a strong genetic divergence existed between the nudibranch specimens and *P. rebus*; the P-distance based on COI gene sequences between them was 0.18 (Supplementary Table S2).

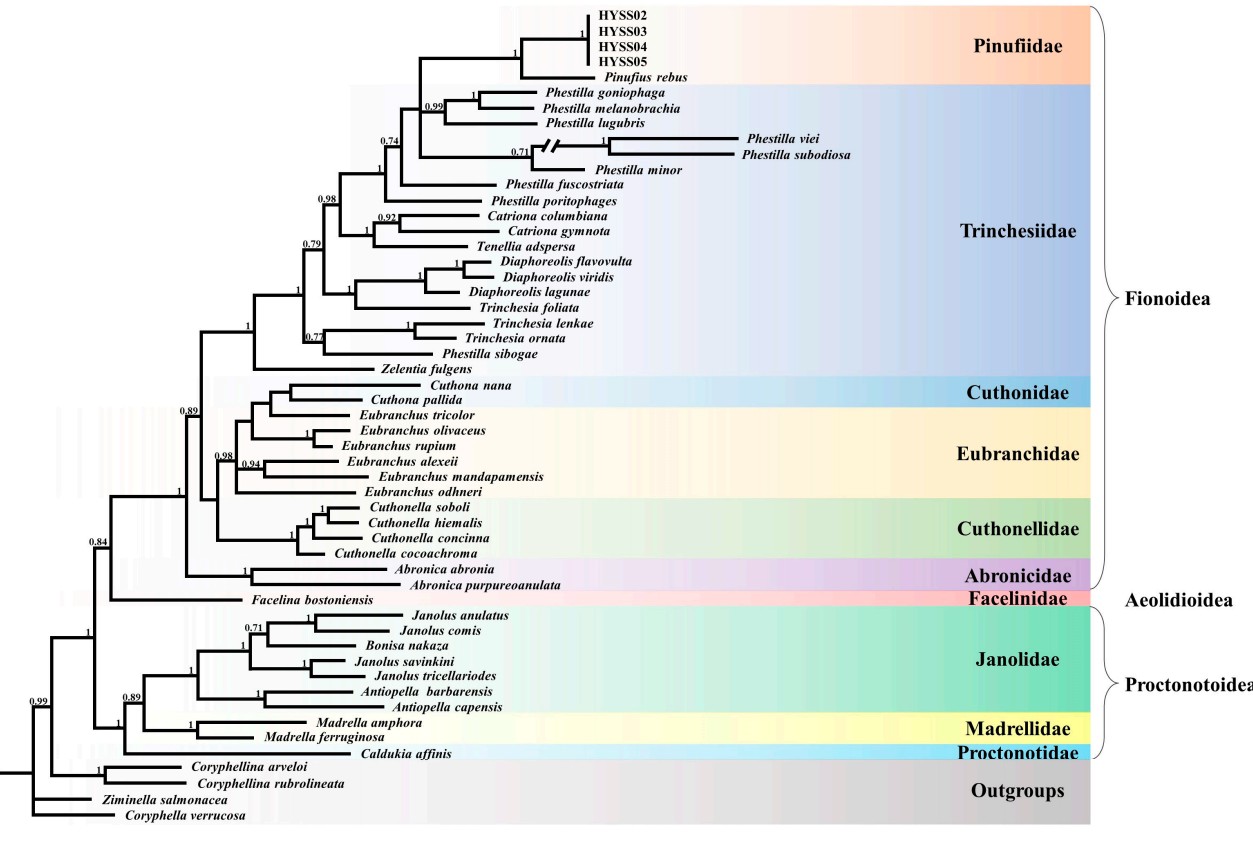

**Figure 7.** Phylogenetic tree based on the concatenated sequences of COI, 16S rRNA and H3 genes, inferred by Bayesian inference (BI) analysis.

## 4. Discussion

In this study, on the basis of morphological comparisons and molecular phylogenetic analyses, we reported a new distribution record for *Pinufius* in the South China Sea and confirmed the taxonomic status of *Pinufius* in the superfamily Fionoidea. Moreover, we put forward some new ideas about species composition and the ecological role of the genus *Pinufius*.

### 4.1. A New Distribution Record for Pinufius in the South China Sea

The genus *Pinufius* has been reported in coral reefs of the Maldives [18], Australia [19], Indonesia [20], and the Philippines [21]. This is the first record of *Pinufius* in the South China Sea.

As Marcus et al. [18] discussed, *Pinufius* is an interesting genus; "it is very aeolid-like in some ways, such as scleractinian parasitism, digestive gland in cerata, and single rachidian tooth in radula; however, in some ways, it is also arminid-like, such as general body shape and arrangement of gut". Our results showed that the external morphological characteristics and feeding habits of the nudibranch specimens in this study were certainly consistent with the original description of *Pinufius* [18,19]. Morphologically, the dorsum of the *Pinufius* sp. in this study was well developed, bearing numerous cerata (Figure 1); these cerata contained digestive glands in which zooxanthellae were present but a nematocyst was absent (Figure 6B). In the anterior, there was a clear oral veil but no oral tentacle (Figure 1), and the radula was uniseriate (Figure 4A). Similarly, the *Pinufius* sp. in this study was remarkably consistent in feeding habits with *P. rebus*, the sole published *Pinufius* species, which also feeds exclusively on *Porites* spp. [19]. Genetic evidence suggested that the clade formed by our four specimens (HYSS02-05) was sister to *P. rebus* with a strong

support (Figure 7). The above results are sufficient to support the conclusion that our specimens belonged to the genus *Pinufius*.

Considering the high dependence of *Pinufius* on *Porites*, as well as the wide distribution of *Porites* in the world [38], we speculated that *Pinufius* could inhabit a range as wide as *Porites*.

### 4.2. The Phylogeny of the Genus Pinufius

Under the superfamily Proctonotoidae, species of the families Janolidae, Madrellidae, and Proctonotidae are typical ceras-bearing arminids [23,24]. Their radulae are multiseriate, and their dorsal cerata contain digestive glands without zooxanthellae [39–42]. *Pinufius* was once classified into the same superfamily with the families Janolidae, Madrellidae, and Proctonotidae because of the similarities among them in shape [24]. However, the cerata and radula of *Pinufius* are very different from those of Proctonotoidea species (Supplementary Table S3). Furthermore, as an important and taxonomically available trait of some nudibranchs [43], their feeding preferences are fundamentally different; the members of Proctonotoidae prey on bryozoans, while *Pinufius* preys exclusively on *Porites* spp. [18,39]. Moreover, the phylogenetic analysis showed that the family Pinufiidae is a distant relative of the families under the superfamily Proctonotidea (Figure 7). In conclusion, our results did not support the previous status of the family Pinufiidae under the traditional ceras-bearing arminid superfamily.

The results sustain the phylogenetic relationship that the family Pinufiidae belongs to the superfamily Fionoidea. Within the superfamily Fionoidea, all *Phestilla* species, with the exception of *Phestilla chaetopterus*, feed on scleractinian corals [27,44,45]. Morphologically, the body of *Phestilla* species is elongated without notum edge, and the cerata contain digestive glands in which zooxanthella are present without nematocysts; the oral tentacles are obvious, and the radula is uniseriate with a formula of 0.1.0 [17,46]. Adult *Phestilla* with limited mobility stay on a coral colony for their entire adult stage [47]. On the basis of the above, it seems that there are many similarities between *Phestilla* and *Pinufius* in their radula, cerata, and the obligate association with scleractinian coral. The close relationship between *Pinufius* and *Phestilla* was also reflected in the present phylogenetic analysis where they formed a well-supported clade together (Figure 7). Fritts-Penniman et al. [20] previously considered that coral-eating nudibranchs had a common evolutionary history, and that *Pinufius* and *Phestilla* should be synonymous. However, the present results are not sufficient to revise the classification of Pinufiidae, because we cannot reasonably explain the significant differences in external morphology between *Pinufius* and *Phestilla*. Some experts considered that morphological synapomorphies possibly occur at earlier larval stages and disappear in the adult stages [17]. This viewpoint has inspired us to check more detailed morphological characteristics of early life history in future studies.

### 4.3. A Suspected New Species of the Genus Pinufius

The present molecular data regarding the COI gene showed that the *Pinufius* sp. in this study formed a sister clade to *P. rebus*, and the P-distance between them (0.18) was much higher than the intraspecific divergence typical of nudibranchs (0.02–0.06) according to prior studies [20]. Morphologically, compared to the descriptions of *P. rebus* by Marcus et al. [18] and Rudman [19], we found some differences on the masticatory border of the jaws. The denticles on the masticatory border of *P. rebus* are bifid with a pair of recurved hooks [18,19], and there was no any denticle on the masticatory border of the *Pinufius* sp. in this study (Figure 4C). Since its first discovery in 1960, *P. rebus* has been the only species in the genus *Pinufius*. In the present study, our specimens were presumed to be a new species in *Pinufius* on the basis of the above morphological differences and molecular evidence. However, further confirmation of the suspected new *Pinufius* species still faces the following impediments:

(1) The reference materials regarding the morphological features of *Pinufius* are limited. The existing textual descriptions and simple sketches cannot provide enough morphological details of the masticatory borders.

(2) The available sequences of *P. rebus* in the NCBI database were collected from the Philippines by Pola et al. [21]. They simply provided the sequences but no morphological depiction of their samples, which restricts us from explaining differences in our molecular results. Moreover, given the inconsistent sampling sites of morphological and molecular data, as well as the hardly perceptible interspecific difference in external morphology, the specimens of Pola et al. [21] may also be from a new species of *Pinufius* differing from *P. rebus* and our specimens. Future studies should focus on the above aspects to confirm the species composition of *Pinufius*.

*4.4. Potential Threats to Local Coral Communities*

In our aquarium, the outbreak of the *Pinufius* sp. caused serious injury to *Porites*. However, there are few reports of massive damage to natural corals by *Pinufius* and other nudibranchs [29]. This may be because there are many natural predators such as carnivorous fish and crustaceans in open water, but less in aquaria. In recent years, with the serious decline in coral reefs around the world, corallivorous predators have also largely been in decline. For example, many studies have indicated that outbreaks of *Acanthaster planci* were due to the overfishing of *Cheilinus undulatus*, *Charonia tritonis*, etc. [48–50]. In Daya Bay, affected by overfishing and habitat destruction, the numbers of coral reef fish and crustaceans have also been in decline [51,52]. Gochfeld et al. [47] highlighted that the outbreak of *Phestilla* might result in changes in coral compositions. Similarly, it is very possible that *Pinufius* sp. would threaten the stability of coral communities.

**5. Conclusions**

In this paper, we described a novel distribution of *Pinufius* in the South China Sea, supporting the systematic status of *Pinufius* under the superfamily Fionoidea, and deduced that the nudibranch in the present study might be a new species of the genus *Pinufius*. The morphological characteristics of this nudibranch are nearly identical with *Pinufius rebus*, including the arminid-like dorsum and oral veil, as well as aeolid-like cerata. However, the masticatory border without denticle is inconsistent with the descriptions of *P. rebus*. Molecular analysis showed that this nudibranch and *P. rebus* were clustered together, and their interspecific genetic divergence was significant. The present study demonstrates that the composition of *Pinufius* is more complex than previously recorded. Morphological and molecular data are needed to verify this new species. Enriching the information of *Pinufius* not only improves the knowledge about this animal but also provides references for relevant research on other corallivorous nudibranchs. Additionally, *Pinufius* and other small corallivorous invertebrates should be taken seriously as potential threats to coral reefs.

**Supplementary Materials:** The following supporting information can be downloaded at https://www.mdpi.com/article/10.3390/d15020226/s1: Table S1. GenBank accession numbers of the sequences used in this study; Table S2. Uncorrected COI p-distances of these species in molecular analysis; Table S3. Summary of the diagnostic characters of related genera [53–59].

**Author Contributions:** Conceptualization, Z.J. and W.N.; methodology, Z.J. and J.X.; formal analysis, Z.X. and B.C.; resources and investigation, P.T. and W.W.; software and data curation, J.X.; writing—original draft preparation, Z.J.; writing—review and editing and supervision, W.N.; funding acquisition, J.X. All authors have read and agreed to the published version of the manuscript.

**Funding:** This research was funded by the National Natural Science Foundation of China (grant numbers 42006098, 42006128, and 42106143), and the Scientific Research Foundation of Third Institute of Oceanography, Ministry of Natural Resources (grant number 2022024).

**Institutional Review Board Statement:** Not applicable.

**Informed Consent Statement:** Not applicable.

**Data Availability Statement:** Publicly available datasets were analyzed in this study. This data can be found at https://www.ncbi.nlm.nih.gov/ accessed on 14 December 2022.

**Acknowledgments:** The authors would like to express their gratitude to Li Gu for her assistance with scanning electron microscope analysis. The authors also thank the reviewers for their constructive and thorough comments that improved the manuscript.

**Conflicts of Interest:** The authors declare no conflict of interest. The funders had no role in the design of the study; in the collection, analyses, or interpretation of data; in the writing of the manuscript; or in the decision to publish the results.

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
