# Peer review of "First Record of Corallivorous Nudibranch Pinufius (Gastropoda: Nudibranchia) in the South China Sea: A Suspected New Species of Pinufius"

_diversity, doi:10.3390/d15020226_

Round 1

Reviewer 1 Report

The paper presents useful new morphological and molecular information on a little studied nudibranch genus that feeds on corals. The relationships of this nudibranch to other coral-feeding nudibranchs will be of interest to biologists in Asia, Australia and the United States. The paper is generally well written, and its arguments are cogently presented and defended. The only major problems with the paper are the very low quality of Figures 6B and C, and 7A, C and D. The SEMs in Fig. 6 illustrate damaged structures wherein it is difficult to see the points being made in the text description. The photographs in Fig. 7 are so small and of such low contrast that it is impossible to understand them or to see what they contribute. They all badly need to be replaced with better material and illustrations.

The following points can be quickly fixed in most cases.

Lines 107 – 110: how as wax removed from the specimens? Other details should be added to this section.

Lines 123 – 125: it is unclear if the authors had the specimens for all of the 49 species,  extracted the DNA and carried out their own sequencing or if the data are from the literature. If the authors did the molecular analyses, it is imperative that Supplementary Table S1 include information on the source of the specimens. Where were they collected? Even if the sequences used are from published sources, data on the collection sites for the specimens should be provided.

Lines 148 – 159: the figures do not appear to support the statement that the cerata are placed in “pairs” –that is four cerata per now. Fig. 2A shows only a single pair.

Fig. 2A. Legend must include definitions of the abbreviations in the figure, e.g., it, ot, ie.

Line 147 and Fig. 2B: the larvae do NOT have cerata. The figure shows a juvenile.

Fig. 3C: the figure illustrates dead embryos disintegrating in their capsules.

Lines 217 – 218. Is this information on Phestilla sibogae taken from the authors’ study or from other published papers? The authors of cited reference 15 state that P. sibogae and P. lugubris are the same species.

Lines 230 – 232: Delete.

Lines 259 and 283; the authors refer to the coral-eating nudibranchs as parasites, when they are clearly predators. Parasites typically live within a single host and cannot live outside. The coral-feeding nudibranchs can easily move between individual coral heads when one is completely devoured.

Line 273: previous is misspelled.

Line 399: there are three more authors for this paper. The senior author, T. M. Gosliner, is the respected authority on nudibranch taxonomy. Have the authors failed to include all author names on other references? I did not check this.

Author Response

Dear Reviewer1:

 Please see the attachment. The revised manuscript is also attached to the bottom of the cover letter. Thank you very much.

Reviewer 2 Report

This manuscript presents very interesting results on range extension in the corallivorous nudibranch Pinufius and the detection of an apparently new species in the genus.  The manuscript is carefully written with good illustrations and generally takes a commendably cautious approach to interpreting the results.  In particular, this comment relates to the authors’ well-explained decision not to formally name the new species at this stage.

In some areas, the authors might be less cautious. For example, the topology of Figure 7 clearly indicates that Pinufiidae is not monophyletic. I realize that this is not the main focus of the manuscript but it may be worth commenting on the doubt that the lack of monophyly raises about the validity of the family. It may also be worth commenting on the aberrant placement of Phesilla sibogae. Is this being driven by one gene only? Is it a case of possible misidentification?

The “deduction” that the range of Pinufius is co-terminal with that of Porites species is referred to twice in the text.  However, there is no information on the true extent of the native range of the new species , or indeed of P. rebus, and this must remain a speculation, albeit that it is worth investigating in the context of the number and range of Pinufius species.

At some point in the text (possibly around the first paragraph of page 2) it would be worth listing the literature on Chinese nudibranchs (and Fionoidea in particular) that the authors have consulted to ensure that Pinufius has not previously been recorded from the South China Sea.

I have made a number of minor comments on the attached annotated file. These are predominantly linguistic.  I would, however, say that the text is very well written. Some care needs to be paid to the references. I have highlighted a few issues but may not have found all instances to which these apply.

Author Response

Dear Reviewer2,

 Our revised manuscript is also included in this file. Please see the attachment.

Thank you a lot.
